# Improving γ-Oryzanol and γ-Aminobutyric Acid Contents in Rice Beverage *Amazake* Produced with Brown, Milled and Germinated Rices

**DOI:** 10.3390/foods12071476

**Published:** 2023-03-31

**Authors:** Ana Castanho, Cristiana Pereira, Manuela Lageiro, Jorge C. Oliveira, Luís M. Cunha, Carla Brites

**Affiliations:** 1GreenUPorto/Inov4Agro, DGAOT, Faculty of Sciences, University of Porto, 4169-007 Porto, Portugal; anavargascastanho@gmail.com (A.C.); lmcunha@fc.up.pt (L.M.C.); 2INIAV—Instituto Nacional de Investigação Agrária e Veterinária, 2780-157 Oeiras, Portugal; cristianapereirags@gmail.com (C.P.); manuela.lageiro@iniav.pt (M.L.); 3GeoBioTec Research Center, Faculdade de Ciências e Tecnologia, Universidade Nova de Lisboa, 2829-516 Caparica, Portugal; 4School of Engineering and Architecture, University College Cork, T12 HW58 Cork, Ireland; j.oliveira@ucc.ie; 5GREEN-IT, ITQB NOVA, 2780-157 Oeiras, Portugal

**Keywords:** γ-oryzanol, γ-aminobutyric acid, α-amylase, *Koji*, *Amazake*, rice bioactives, response surface methodology

## Abstract

Rice is an important source of γ-oryzanol (GO) and γ-aminobutyric acid (GABA), which are bioactive compounds that may benefit blood lipid and pressure control. Both GO and GABA can be improved by germination and fermentation. Fermentation with *A. oryzae* produces *Koji*, a rice-based starter for *Amazake*, a naturally sweet beverage. Germinated rice (brown and milled rice), were tested to improve those bioactive compounds during the fermentation process. The resulting *Koji* was optimised to GO and GABA through a response surface methodology; α-amylase activity and starch content were also assessed. The different rice matrix resulting from the germination largely impacted the biosynthesis of GABA, α-amylase and starch contents. *Amazake*, obtained by germinated rice, has increased GO and GABA contents when compared to the one obtained from milled rice (from a non-detectable value to 27.65 ± 0.23 mg/100 g for GO and from 163.95 ± 24.7 to 271.53 ± 5.7 mg/100 g for GABA). A panel of 136 Portuguese consumers tasted the beverage in a blind overall tasting test followed by an informed test, using 9-point scales. The consumer scores had a mean value of 4.67 ± 1.9 and 4.9 ± 1.8, meaning that cultural differences may play an important role with regard to liking and accepting *Amazake*.

## 1. Introduction

Rice is considered a staple food in many countries, particularly in Asia. Despite the low consumption rate in Europe, Portugal is the largest rice consumer, with innumerable ways of consuming rice in its gastronomic tradition. However, rice is not consumed at its full nutritional potential, since the bran and germ are discarded with the industrial processing, and rice is consumed after milling [1]. In recent years, the nutritional compounds of whole rice and rice bran have been studied to explore natural ways of increasing its content in rice-based foods. One of those compounds is γ-oryzanol (GO) [2,3]. GO is a bioactive compound known for its potential to regulate lipid metabolism, reducing the LDL (Low-Density Lipoprotein) and total cholesterol [4,5,6,7]. GO is synthesised during the maturation of the rice grain, being responsible for different regulatory functions, properties and structures; therefore, the GO content is not only affected by genetic factors but also by climate and growing conditions [8,9,10].

Another important compound in rice bran is γ-aminobutyric acid (GABA), a nonprotein amino acid. GABA is an important neurotransmitter responsible for lowering the nervous system’s activity, effectively lowering blood pressure [11,12,13,14] and treating epilepsy [15]. GABA is mainly synthesised by biochemical processes such as germination and fermentation. During those processes, the α-ketoglutarate (contained in large quantities in the bran) is transaminated to glutamic acid, which is decarboxylated to GABA through glutamic acid decarboxylase (GAD) [12,13]. Thus, increased levels of GABA can be obtained by reinforcing the germination status of the grain. The amount of GABA in rice bran ranges from 10.7–58.0 mg/100 g (before germination) and 90.0–350.0 mg/100 g (after 10–12 h of rice germination) [16].

The germination and fermentation processes have been employed empirically to increase nutritional value and enhance the assimilability and palatability of foods. The most important microorganism employed in rice fermentation is *Aspergillus oryzae*, which was named after its occurrence in rice (*Oryza sativa*) [17]. *A. oryzae* has been used in Asia, mainly in Japan, as a starter for a solid-state fermentation named *Koji*, which is the basis of many traditional products, including soy sauce, miso, sake, and *Amazake* [17,18]. Solid-state fermentation is a low moisture fermentation, providing just enough water to support the growth of filamentous fungi’s penetrative and aerial hyphae, producing enzymes that would not typically be produced during other fermentation processes [19,20]. Although *A. oryzae* can ferment other products, rice is the most used food substrate for *Koji* production. The process to obtain *Koji* can be seen as a system which includes mainly (1) time, (2) temperature, and (3) moisture [21]. Ventilation, substrate type, pH, and the initial quantity of the inoculum are also essential factors in *Koji*-making [20,22]. *Amazake* is the simplest way of using *Koji*. *Amazake* is a non-alcoholic, white-coloured, sweet drink that can be made only by mixing *Koji* with water or by adding steamed rice. In the processing of *Amazake*, *Koji* acts as a starter for starch hydrolysis, where their α-amylases break the rice starch into glucose. Therefore, besides *Koji*’s unique flavour, *Amazake* is naturally sweet (without added sugars); contrary to other sweet beverages, which contain sucrose or fructose, *Amazake*’s sweetness is mainly realized through naturally produced glucose [23,24]. This is an advantage, since the final glucose content of *Amazake* can be controlled and adjusted through a longer or shorter rice *Koji* fermentation time [25].

Despite being closely related to the *Aspergillus flavus* species, *A. oryzae* is considered a safe organism by the World Health Organisation [26], if the microbiological safety of *Koji* and its products are produced with approved starter cultures, appropriate rice *Koji* fermentation processes, and storage conditions are used [20]. The safety of *A. oryzae* arises from the fact that it is unable to produce aflatoxins, as the genes responsible for its biosynthesis are absent or dysfunctional [18,20,23,24,27]. *Koji Amazake* can also be considered microbiologically safe, as starch is hydrolysed at a temperature range where harmful bacteria do not grow [24].

Few authors have studied the impact of *Amazake* intake on human health. Kurahashi and Yonei [23] studied the excess intake of *Amazake* (for four consecutive weeks) in 24 healthy subjects with high blood glucose levels. The body mass index (BMI) of the subjects did not change, but LDL cholesterol and glycated haemoglobin (HbA1c), a marker for diabetes, had a significant slight decrease, returning to the typical values after the test; the excess intake of *Amazake* led to a significant decrease in blood pressure. The authors related the hypotensive effect with the presence of GABA. Kikushima et al. [28], in a randomised, double-blind, placebo-controlled trial, evaluated the effects of *Koji Amazake*, finding that the daily intake of *Amazake* decreased systolic blood pressure. Other authors [25,29] studied the impact of *Amazake* on other health issues with positive results.

This work aimed to develop a rice-based beverage (*Amazake*) via a germination and rice *Koji* fermentation process with *A. oryzae*. As both GABA and GO can be improved naturally through biochemical processes, this work explores ways of increasing *Amazake’s* GABA and GO contents, improving *Amazake’s* nutritional value and evaluating its acceptability by Portuguese consumers.

## 2. Materials and Methods

A schematic view of the experimental methodology can be observed in Figure 1. Different rice matrices were selected for this study: milled, whole and germinated rice; germination rice was subjected to a pre-selection (through the employment of three different germination times) to obtain the greatest GABA and GO contents.

The three raw samples were further soaked, cooked and inoculated with *A. oryzae* to produce *Koji*. The GABA content and α-amylase activity were analysed in the *Koji* preparation process to optimise the best inoculation time and temperature.

The optimised *Koji* was subjected to a starch hydrolysis process to obtain *Amazake*. The sweetness and texture parameters were monitored through the starch hydrolysis process to reach similarity with commercial *Amazake* (fermented brown rice *Amazake* from Kenshô, Spain) in terms of sweetness. For texture, the used reference was a commercial yoghurt beverage (plain liquid yoghurt from a major Portuguese retailer brand).

The improved *Amazake* was then presented to consumers to assess its overall likeability, and it was also evaluated for acceptance after elicitation with a final product prototype.

### 2.1. Sample Selection

Commercial milled and brown rice from the variety Ariete (from Orivarzea SA, Portugal) were acquired in the market and kept sealed at room temperature until use.

Germinated rice was prepared from brown rice, washed three times, and submerged in distilled water for 24 h; the samples were kept at 25 °C for 48 and 72 h, being washed every 4 h to avoid fermentation. Samples of 50 g were collected and freeze-dried. For comparison purposes, the raw brown rice sample was considered the control, and the subsequent samples were identified as 24 h, 48 h and 72 h germinated rice. The GABA and GO data were acquired, and the sample with greater values of those compounds was selected for further processing.

All of the samples were analysed in triplicate.

#### 2.1.1. GABA Extraction and Quantification

GABA extraction and quantification were performed according to Jannoey et al. [30], with modifications. The freeze-dried samples were extracted with 70% ethanol. The samples were dissolved with 2 mL of ethanol, mixed in a vortex, shaken for 30 min at room temperature, and centrifuged at 4 °C for 10 min at 10,000× *g* (Laborzentrifugen 2K15, Sigma, Osterode am Harz, Germany). The supernatant was collected, and the procedure was repeated two more times, until collecting a total of 6 mL extract. The derivatisation was performed with 2-hydroxynaphthaldehyde (HN), by adding 1 mL of the extracted samples to 0.5 mL HN-methanol solution (0.3%) and 0.5 mL of borate buffer (pH 8), incubating at 80 °C during 15 min in a water-bath and cooling to room temperature. The resulting extract was filtered through a 0.22 µm nylon syringe filter (FilterTECH, Saran, France) before injection.

The quantification of GABA was performed by reverse-phase high-performance liquid chromatography (RP-HPLC) using Waters 2695 series equipment (Waters, Milford, MA, USA) connected to a diode array detector (DAD, Waters 2996, Waters, Milford, MA, USA). The amino acid HN derivative separation was performed in a C18 column (Sunfire 5 µm, 4.6 × 250 mm, Waters, Milford, MA, USA), which remained at a temperature of 40 °C (Waters column thermostat Jetstream 2 plus, Waters, Milford, MA, USA). The mobile phase used was methanol (A): water (B), with the following gradient: 0–4 min: 20–50%(A); 4–6 min: 50–80% (A); 6–8 min: 80–100% (A); 8–10 min: 100% (A) and from 10–14 min: 100–20% (A), at a flow of 0.8 mL min^−1^ and with an injection volume of 20 µL, with a total analysis time of 20 min.

The GABA-HN derivative identification was performed at 231 nm by comparison of the GABA-HN peak retention times from the GABA-HN spectra. Sample fortification with GABA standard was undertaken for all samples prior to HN derivatisation, and the results of the fortified samples were compared with non-fortified samples to validate the GABA identification. The GABA quantification was based on an external calibration curve made with GABA standard aqueous solutions (10 to 1000 mg/L). A linear regression was obtained between the GABA-HN peak area and the GABA content in mg/mL (total GABA-HN area = 144,448,962 [GABA] + 5,176,153) with a determination coefficient of 0.995. GABA quantification was measured in mg of GABA per 100 g of sample, and all samples were extracted in duplicate.

#### 2.1.2. GO Extraction and Quantification

The γ-oryzanol (GO) quantification was performed by RP-HPLC equipped with a DAD using the same HPLC system described for GABA quantification using a Waters Spherisorb ODS 2 separation column (4.6 × 250 mm, 5 µm, Waters, Milford, MA, USA), according to the method reported by Lageiro et al. [8], with minor modifications. The extraction of GO was carried out from freeze-dried *Koji* lipid residue according to the procedures reported by Castanho, et al. [1]. The GO identification was made at 325 nm by retention times and spectra comparisons with GO standards from TCI Europe. The quantification was based on an external calibration curve with GO standard solutions (10 to 900 mg/L). A linear regression between the total GO area (peak area sum of the different GO compounds’ peak areas) and the GO content in mg/mL was obtained (total GO area = 29.10 × [GO] 58.99), with a determination coefficient of 0.9995. GO quantification was measured in mg of GO per 100 g of sample, and all samples were extracted in duplicate.

### 2.2. Rice Cooking and Inoculation

Rice was cooked according to Saigusa and Ohba [31], with few modifications. The milled rice (250 g) was washed three times and soaked in distilled water for 20 min. It was drained for 15 min, and the water content was adjusted to achieve 118% of the initial raw rice weight with distilled water. The rice was then steamed for 40 min in a rice cooker steamer (CKSTRC4723-050, Oster 600 mL, Sunbeam Products, Inc., Boca Raton, FL, USA), wrapped in 20 × 20 cm non-woven pads (Wells, Portugal). The cooked rice was allowed to rest for 10 min while wrapped. After removing the cloth, the rice was allowed to cool in a tray until the temperature decreased to 40 °C. For brown rice, the cooking method was slightly modified: the raw brown rice was soaked for 5 h to adjust the water content to 118% of the initial raw rice weight and cooked for 60 min. The procedure was the same for germinated rice apart from the initial water correction, as the water uptake was given by the germination time.

The water content of all samples was adjusted (133% raw weight) with distilled water to ensure process standardisation before inoculation. The cooked rice was then inoculated with *A. oryzae* spores obtained from Kenshô (Spain) according to the producer’s instructions (1 g/kg of raw rice).

### 2.3. Koji Preparation and Optimisation

A circumscribed central composite design using rice *Koji* fermentation time and temperature as independent variables was used to obtain a surface response, with the values presented in Table 1. Ten experimental combinations were determined with two levels (−1, +1), two repetitions of the central point (0), and two levels of axial points (−α, +α) [32]. The experimental conditions were defined as between 20 and 60 h and 32 and 42 °C, according to Narahara [33]. The selected points were used for GABA, α-amylase activity, and starch content evaluation.

Before analysis, the preparation of *Koji* samples was based on Hong and Kim [34] and previous pilot studies from the research group. Fresh collected *Koji*, 20 g, was mixed with 60 mL of distilled water and homogenised in an Ultraturax (T25, IKA, Germany). From that mixture, 6 mL were diluted in 16 mL of distilled water and centrifuged at 10,000× *g*; the remaining *Koji* mixture was freeze-dried, ground to flour, passed through a 120 µm sieve, and stored in dry conditions for subsequent analysis, as *Koji* extracts. All the analyses were treated as independent triplicates.

#### 2.3.1. α-Amylase Activity Assessment

The α-amylase activity of *Koji* extracts was measured using the Ceralpha^®^ assay procedure (AOAC Official Method 2002.01) by using benzylidene-blocked *p*-nitrophenyl maltoheptaoside in the presence of thermostable α-glucosidase. The activity was calculated according to the Ceralpha^®^ assay procedure using the Megazyme kit calculator for Microsoft Excel, and expressed as Ceralpha U per g *Koji*, where one Unit of enzyme activity is the amount of enzyme releasing one μmole of *p*-nitrophenol per minute under the defined assay conditions.

#### 2.3.2. Starch Measurement

Starch data was collected using NIR transflection MPA equipment (Bruker Optics, Germany), according to Sampaio, et al. [35]. The data was obtained using the B-FING cereals calibration model (Bruker Optics, Germany).

### 2.4. Amazake Selection and Preparation Processes

*Amazake* was produced by hydrolysing the optimised *Koji* by adding water at a specific temperature and time. The suspension viscosity was measured, following each water addition, until the target value (plain liquid yoghurt, 0.285 Pa.s) was reached. As the target texture was achieved, the sweetness was determined by °Brix, through time, to achieve commercial *Amazake* sweetness (fermented brown rice *Amazake*, 21%).

The *Amazake* was mixed in a cooking robot (Thermomix TM6, Vorwerk, Germany) at the highest speed in order to homogenise the beverage.

#### 2.4.1. Starch Hydrolysis and Viscosity Measurement

Starch hydrolysis was performed at a small scale in a Rapid Visco Analyser (RVA 4800, Perten Instruments, Sweden) at 50 °C at the constant speed of 1000 rpm for 30 min at different *Koji*:water proportions (1:1; 1:1.5; 1.75 and 1:2). The samples were cooled down to 4 °C, and the viscosity after cooling was measured at the constant speed of 1000 rpm for 2 min.

#### 2.4.2. Brix Measurement

The sweetness of the *Amazake* was measured by its Brix (%) in a refractometer (PR-201, Atago, Japan) after filtering the suspension through a 45 µm syringe filter. The measurements were performed after 10, 15, 25 and 35 min.

### 2.5. Consumer Evaluation

A total of 111 consumers evaluated the *Amazake* beverage at an open science fair, which included students and families, representing a broad demographic group. In the first stage, the consumers tested the product without any information other than that it was a new and experimental rice beverage. The overall liking was evaluated using a 9-point scale, ranging from 1—“dislike extremely” to 9—“like extremely” [36].

In the second stage, after the first questionnaire, a prototype of the product was created, including a bottle full of “DIS! *Amasake*” with a label containing nutritional information, due date, ingredients, and logotype, and a flyer with information about the potential health benefits as well as information about the objectives of the project. The consumers filled out the Food Action Rating Scale (FACT) developed by Schutz [37], that was adapted for the product. The 9-point FACT scale was: 1—“I would drink this only if forced“, 2—“I would drink this if there were no other food choice“, 3—“I would hardly ever drink this“, 4—“I do not like this but would drink this on an occasion“, 5—“I would drink this if available but would not go out of my way“, 6—“I like this and would drink it now and then“, 7—“I would frequently drink this“, 8—“I would drink this very often“, 9—“I would drink this every opportunity that I have“. Both the liking and FACT scales were translated into Portuguese, according to Ribeiro, et al. [38]. Demographic data (age, gender and education level) and free comments about the product were also collected with the overall liking data. Free comments have been analysed by grouping terms into categories and sub-categories derived from the content analysis [39] following a triangulation procedure [40]. All participants were aged 16 years old or above, and were willing to participate. Following the Helsinki statement, an informed consent was given, and participants were assured that all private information would remain anonymous. All minors (<18 years old) were accompanied by their consenting parents or educators. The research team has enforced procedures to guarantee adherence to the European General Data Protection Regulations.

### 2.6. Statistical Analysis

To analyse data from the rice *Koji* fermentation process, following the Central Composite Design, a quadratic model (see Equation (1)) was fitted to GABA, α-amylase activity, and starch content data, where b_0_ represents the constant term, b_1_ and b_2_ the linear terms, b_3_ and b_5_ the quadratic terms, and b_4_ the interaction term. *x*_1_ represents the rice *Koji* fermentation temperature (°C), and *x*_2_ represents rice *Koji* fermentation time (h).
(1)$$y=b_{0}+b_{1}x_{1}+b_{2}x_{2}+b_{3}x_{1}{}^{2}+b_{4}x_{1}x_{2}+b_{5}x_{2}{}^{2}$$

Raw data inspection was performed in advance, and outliers were identified among triplicates at each sampling point, following the Grubbs method (Appendix A). A linear model building approach was implemented through a stepwise regression approach, minimising Akaike’s AIC parameter [41]. Overall model fitting was assessed by the adjusted coefficient of determination R_adj_^2^ [32].

The comparison of processing conditions at the additional steps, such as the impact of the germination time on GO and GABA, was performed using a one-way ANOVA, followed by Tukey’s post hoc multiple comparison test, if applicable. All tests were performed at a 95% confidence level using XLStat for Microsoft Excel.

## 3. Results and Discussion

### 3.1. Sample Selection

The rice bran proteins, lipids, and mineral content are considered undesirable for that specific fermentation process, and traditional rice *Koji* is produced with milled rice [21], so a milled rice sample was included as a control in the improvement of GO and GABA contents; a brown rice sample was also included for presenting a naturally richer content of those compounds due to the presence of the bran [1,42].

As GO and GABA contents also increase with germination, a previous study was conducted to select the best germination times. Figure 2 shows the contents of GO and GABA over germination time.

Both GO and GABA presented an increase in their content after 24 h of immersed germination when compared to raw rice, from 23.09 ± 1.193 to 41.17 ± 0.835 mg/100 g and 57.783 ± 3.60 to 67.73 ± 0.22 mg/100 g, respectively (Figure 2). After 24 h, the GO content lowered, and the GABA content increased; however, those changes were not statistically significant (*p* < 0.05). Wu, et al. [42] also reported an increase of the GO in the first germination stage, followed by a slight decrease; however, regarding GABA, the authors denoted a constant increase during germination. Wu, et al. [42] attributed the enhancement of GO in the first stages to the natural presence of lipases, promoting the decomposition of various lipid substances, including GO. Munarko, et al. [43] tested the effect of the germination process on the GABA content in four different rice varieties, reporting differences in the increase rate of GABA content with the germination time; the same authors also reported differences between GO content changing rates between the four varieties along germination time. The increase of GO content with germination is also reported by other authors [44].

Considering the results obtained, a germinated brown rice sample, after 24 h of submersion, was selected for the fermentation trials to obtain rice *Koji,* in addition to the milled and brown rice samples.

### 3.2. Rice Preparation

As reported previously, time, temperature, and moisture are the most critical factors influencing *Koji* making [21]. GO and GABA content can be affected by the processes carried out during *Koji* making. Besides temperature and time, the system’s moisture is also essential. The system’s moisture combines the media’s humidity and cooked rice’s moisture. While initially the moisture content is about 95–98%, during the rice *Koji* fermentation process, the moisture is lost due to fungal metabolism [24], therefore being essential to cover the *Koji* with a cloth. The system’s moisture was standardised by carefully monitoring the water content based on the raw weight of the rice.

The gelatinisation stage of starch, from β-starch to α-starch, is important in *Koji* making, as complete gelatinization is needed to allow the tip of the *A. oryzae* hyphae to extend forward into the rice grain and not only over the surface, thus producing more desirable compounds [24]. The gelatinisation state of the rice was also verified before inoculation.

According to Gomi [17], the optimum temperature for *Koji* making is between 32–40 °C. As *A. oryzae* cannot grow above 44 °C, the cooked rice was cooled to 40–42 °C before inoculation.

### 3.3. Optimisation of Koji According to GABA and α-Amylase Activity

Temperature and time were the selected variables to optimise the *Koji* making:

(1) Temperature can affect the production of metabolic compounds such as GABA, and can also dictate the type of compounds produced (e.g., temperatures up to 37 °C are reported to increase the production of protease, while higher temperatures increase amylase activity), depending on the strain specificities [21,22]; however, fermentation temperature depends not only on the external specified parameters, but also on the exothermic reaction that occurs during that stage (rising the temperature to 40–42 °C); therefore, it is necessary to mix *Koji* and to cool it and ventilate it, especially when working with large batches [24].

(2) Time is also crucial in *Koji* making, as it defines the stage of growth;

(a) spores start germinating three to five hours after attachment to the matrix;

(b) at 20 h the spores grow on the grain surface; and

(c) the hyphae grow, and the mould extends its hyphae into rice grains, spreading over the matrix at 44 h, secreting high molecular weight hydrolytic enzymes (e.g., amylase and proteases) through the process [18,24].

During this stage, GABA and α-amylase activity were selected as the target parameters to measure, as GO content is related to lipid metabolism and is not expected to change during rice *Koji* fermentation, and α-amylase will break the starch in the following hydrolysis process, therefore being essential to *Amazake* production. Starch, the main compound of rice, was measured to observe the changes occurring during the germination and fermentation processes.

Table 2 and Figure 3 present the regression models’ parameters and model fit for GABA, α-amylase activity and starch content, and the contour plots for those models.

Germination is a process that occurs in three phases [45]:

(1) The seeds become fully hydrated;

(2) Activation of the metabolism to mobilise nutrients to grow the radicle, which emerges at the end of this phase;

(3) The seed absorbs even more water, and there is a significant mobilisation of reserve material (i.e., carbohydrates which are hydrolysed and metabolised, leading to the growing of the seed).

During the germination process, the rice matrix suffers irreversible changes: α-amylase and α-glucosidases are synthesised; thus, the starch content decreases due to the partial hydrolysis of starch [46]; the peptidase activity levels increase [47], leading to protein hydrolysis, which is not only degraded into peptides but also into free amino acids [45]. Given these changes, the GABA synthesis in the germinated rice during fermentation is carried out in a completely new matrix; if, on the one hand, the free glutamic acid present in raw rice bran was consumed as a substrate for producing GABA during the germination, on the other hand, during rice *Koji* fermentation, the enzymes produced by *A. oryzae* can break proteins leading to the existence of more free glutamic acid that is latterly synthesised to GABA. However, as *A. oryzae* produces proteases at a lower temperature, there is a need to decrease the fermentation temperature; at the same time, the time factor is also important to allow the action of the proteases, which may explain the differences between the predicted time to achieve the maximum values of GABA regarding brown and milled rice, and the germinated rice in Figure 3.

The α-amylase predicted time and temperature for maximum α-amylase activity also follows the pattern of GABA synthesis (Figure 3). In milled rice and brown rice samples, α-amylase is formed at the time/temperature suggested in the literature [21,22]; in the germinated rice matrix, α-amylase reaches its more significant activity at the lower temperature and maximum time. Here, the hypothesis presented before may also apply, and as the matrix changes, the proteases formed at lower temperatures may also change, freeing some compounds that promote the increase of α-amylase activity. Despite the differences in the initial starch amounts of milled and brown rice (82.77 ± 1.410 and 73.05 ± 0.351 g/100 g, respectively), the starch that remained after fermentation was similar in both matrices. Germinated rice predicted starch content shows a lower amount of starch at lower temperatures and maximum time, meaning that the starch structure may be responsible for those differences (Figure 3).

Table 3 shows the maximum concentrations of GABA and α-amylase activity, according to the simultaneous linear optimisation of both the GABA and α-amylase activity models. The results show the similarities between milled and brown rice regarding time, temperature, and maximum predicted values. Germinated rice requires more time and less temperature to obtain higher values of GABA than the other matrices; however, despite the maximum α-amylase activity occurring at the same time and temperature, the values are very low compared to the other matrices. Despite the low α-amylase activity, germinated rice was selected to produce the *Amazake* beverage, as pilot studies showed that germinated rice α-amylase could produce satisfactory results on the enzymatic hydrolysis.

The predicted results were confirmed by a GABA and α-amylase activity analysis, showing results near the expected: regarding GABA, a concentration of 251.52 ± 6.3 mg/100 g, and regarding α-amylase activity, 4.04 ± 0.07 CU/g. The starch content was also assessed in the selected *Koji*, presenting a predicted value of 55.89 g/100g, and a response of 59.11 ± 0.16 g/100 g.

### 3.4. Amazake Preparation

*Amazake*, a beverage naturally sweetened by the action of enzymes, can be produced by adding *Koji* and water and exposing it to a controlled temperature over a specific period of time. While the water addition influences texture, it will also change the concentration of α-amylase in the system, manipulating the sweetness of *Amazake*. Time is also a key factor, as sweetness can be controlled by the time the enzymes take to hydrolyse the starch to glucose [48]. The starch enzymatic hydrolysis is carried out in two stages: dextrinisation (or liquefaction) and saccharification. During dextrinisation, there is a breakdown of starch in oligosaccharides, polysaccharides, or maltodextrin with a loss in viscosity; in the saccharification phase, the maltodextrins are mainly converted into glucose [48,49]. Figure 4 shows the °Brix and viscosity values progression over time in the same *Koji*:water samples subjected to 50 °C in RVA, where the rapid decrease of viscosity represents the liquefaction phase, and the slower increase of °Brix represents the saccharification stage.

The texture and sweetness of *Amazake* were first tested on a small scale using an RVA at a constant temperature and speed. The optimum temperature for *A. oryzae* α-amylase to act on the substrate is reported to be 50–60 °C [24]; due to the reported low α-amylase activity, the system temperature was maintained at 50 °C. The texture was first improved to reach the yoghurt viscosity (0.285 Pa.s). As expected, the viscosity decreased with the water addition (Figure 4A). As the 1:1 *Koji*:water concentration showed a viscosity similar to the commercial yoghurt (Figure 4B), it was selected for the *Amazake* production at a larger scale for sensory evaluation.

The selected *Koji*:water concentration was then subjected to starch hydrolysis over time, also using the RVA, to obtain the °Brix value. Although °Brix is not a very accurate measure, in this case it was very useful, as a rapid measurement was preferred to a more accurate one due to the rapid action of α-amylase, as it could modify the results in a short amount of time. *Amazake* °Brix was set considering the reference of the commercial *Koji* °Brix, as the other compounds influence the sweet flavour in the *Amazake*, namely volatile components that are largely dependent on lipid oxidation in the *Koji* fermentation stage [50]. Therefore, instead of 14.3 °Brix of yoghurt, a 21 °Brix of commercial *Koji* was considered.

The *Amazake*, produced in the large batches (2 L) for consumers’ sensory evaluation, had a relatively small change in the texture (from 0.255 ± 0.007 to 0.345 ± 0.035 Pa.s); in the large batches, the starch hydrolysis was carried out over five hours, with hourly °Brix measurements.

### 3.5. Effect of Germination and Rice Koji Fermentation on GO and GABA of Rice after Cooking

Figure 5 compares the effects of *Amazake*’s processing stages (cooking, germination, fermentation and starch hydrolysis) on GO and GABA contents with the traditional milled rice *Amazake*. As expected, steam cooking did not affect GO and GABA content, since both are resistant to boiling temperature [51,52]. With germination, both compounds increased, as was already reported in Figure 2; however, while for GO the value was 78.3% higher, for GABA the increase was not significant (*p* > 0.05), and represented only 17.2%.

Despite the higher GO content in germinated rice when raw, the amount decreased after cooking, which can be related to any leaching of the compounds; the differences between the effect of boiling in raw and germinated rice may be related to the different matrix that germination generates. GO values increased with rice *Koji* fermentation and even more when germination and fermentation were used, as well as in the optimised time × temperature samples (37 °C/20 h).

Regarding GABA, the results show that fermentation greatly impacted the concentration of the compound: non-germinated brown rice fermented at 37 °C for 40 h presents a GABA value of 249.95 ± 24.31 mg/100g; when fermented after germination, the content of GABA decreases; however, after optimisation, (32 °C for 60 h), the GABA content increased to the values of the non-germinated brown rice. The reported values may be explained by starch availability, reported in Table 2 and Figure 3, as *A. oryzae* fermentation occurs based on carbohydrate metabolism [17,18], the production of compounds may be compromised or, in this case, takes longer, derived from the changes occurred during the germination process. Wang, et al. [53] studied the effect of germination and fermentation on GO and GABA contents in rice inoculated with *Bacillus subtilis Natto.* The authors reported a significant decrease of GO and GABA after fermentation in both germinated and non-germinated rice, which is understandable as the starter cultures are different, thus having different metabolisms.

Figure 5 also shows that the GO content decreases sharply after starch hydrolysis, which may be related to the action of other enzymes (e.g., lipases that are also formed during fermentation [19,20]). Regarding GABA, there is a slight content increase after hydrolysis. When comparing the optimised *Amazake* with the one made with milled rice *Koji* under traditional conditions (37 °C/40 h), the optimisation process improved GO and GABA contents.

### 3.6. Consumer Evaluations

The final *Amazake* product was presented to the consumers at an open science fair. At first, no information was available about the product, and the consumer tried and evaluated the product hedonically. Secondly, the product was shown bottled under the brand DIS!Amasake with a full label and according to the EU labelling legislation for food products [54].

Although the Portuguese gastronomic tradition is rich in rice meals and even deserts [55], eating germinated or fermented rice is not a habit, and to our knowledge, there are no industries in the country producing those products. Despite this, it is possible to find *Koji* products in the market, although only in niche shops or online. Thus, the *Koji* flavour is unknown to most Portuguese consumers.

Overall liking, before elicitation, had a relatively low score, as 42.3% of the participants rated the beverage in the lower range of the scale, and only 37.8% rated the beverage in the higher range (Figure 6). The median and mode score was five, meaning that the product was neither liked nor disliked. The observed values may be related to various factors, including the novelty of the product, and/or cultural differences, as *Amazake* is a traditional drink in Japan. Various authors [56,57,58,59,60] have reported how cultural differences impact food preferences in a variety of cross-cultural studies. Prescott et al. [60] found that Japanese subjects had a higher preference for acidic and umami flavours at higher concentrations than Australian subjects. The Japanese beverage *Amazake* presents an acidic flavour due to the compounds formed naturally in the fermentation process.

Forty consumers filled out the optional open comments section, resulting in a total of 47 different entries. The data were grouped into four categories: taste (*n* = 23), texture (*n* = 11), aroma (*n* = 4), and general attributes (*n* = 9). The prominent taste characteristics were “bitter” (*n* = 5), “strong” (*n* = 4) and “yeast” (*n* = 4); however, those aspects are not correlated to the liking parameters, but rather to the individual preferences and habits. Regarding texture, the most reported characteristic was “thick” (*n* = 7), which may be related to the expectation of the testers given their knowledge of rice beverages in the market, which are texturally similar to milk. The aroma was described as “unpleasant” (*n* = 4), a preference parameter. The participants who described the beverage regarding general attributes reported it as “different/unknown/unexpected” (*n* = 6), and a participant commented that it “requires habituation“. This last comment is interesting, as the literature reports that food preferences are acquired over time [61]; liking a substance tends to increase with familiarity and exposure.

Beverage acceptance after exposure to the prototype and related nutritional benefits shows a slight increase when compared with the overall liking scores (Figure 6). Thus, when comparing the differences between liking and acceptance, the participants who most disliked the product were positively impacted by the nutritional elicitation, and the participants with higher liking scores showed a more dispersed attitude; however, the overall liking scores were positively correlated with the FACT scores (R = 0.743).

As for overall liking scores, the impact of nutritional elicitation can be influenced by cultural aspects [62]. When comparing cross-cultural applications of the Food Choice Questionnaire, Cunha, et al. [63] showed that there is a visible difference between the food choice determinants of Japanese and Portuguese consumers: for the Japanese, the health and natural content are more important than sensorial traits, while for the Portuguese the sensorial traits are the most important. Therefore, Portuguese consumers may be less prone to accept unknown health-based foods with a perceived lower sensory appeal.

## 4. Conclusions and Future Work

The results show that all the measured compounds are affected by the rice raw material (germinated, brown or milled), particularly regarding germinated rice, and also by further fermentation. The germinated rice *Koji* was selected for the *Amazake* preparation due to its higher γ-oryzanol (GO) and γ-aminobutyric acid (GABA) contents. In the selected sample, the GO concentration increased with the germination process in raw samples (from 23.09 ± 1.19 to 41.17 ± 0.84 mg/100 g), and after fermentation (to 44.17 ± 1.8 mg/100 g); however, after the starch hydrolysis process, the values of GO decreased to 27.65 ± 0.23 mg/100 g. The use of the traditional milled rice as raw material for *Amazake* presented a non-detectable GO, while significant GO amounts can be obtained using germinated rice. The GABA content in the selected sample increased with fermentation alone from 57.783 ± 3.60 to 249.95 ± 24.31 mg/100 g, and for the predicted optimum conditions (271.53 ± 5.7 mg/100 g), maintaining the value after starch hydrolysis. Compared to the GABA content of the milled rice *Amazake* (163.95 ± 24.7 mg/100 g), it also presented a great increment. Despite the nutritional improvement associated with the bioactive compounds obtained in the *Amazake*, when presented to the consumers, the overall liking scores were low, which can be attributed to the product novelty and low familiarity with the concept of fermented rice beverages for Portuguese consumers. Nevertheless, as there is room for improvement and even adaptation of the formulation to the Portuguese consumers’ habits, further sensory improvement may be sought in the future with different variants of the fermented beverage, as well as the evaluation of acceptance within different consumption contexts.

## Figures and Tables

**Figure 1 foods-12-01476-f001:**
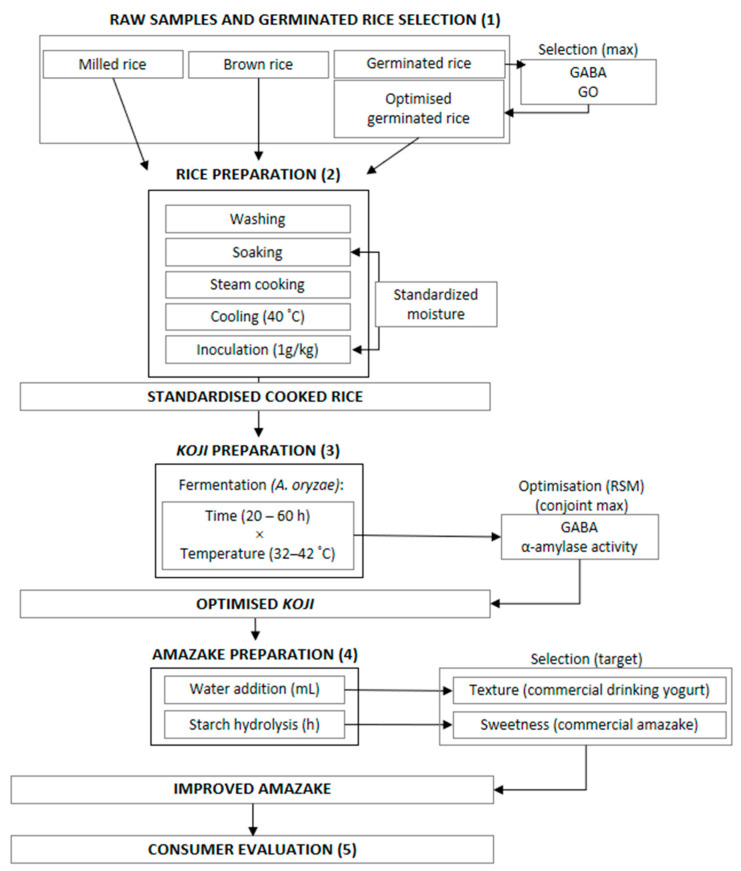
Schematic view of the experimental methodology, with numbers representing materials and methods, as well as discussion sections (e.g., 1 represents Section 2.1 and Section 3.1).

**Figure 2 foods-12-01476-f002:**
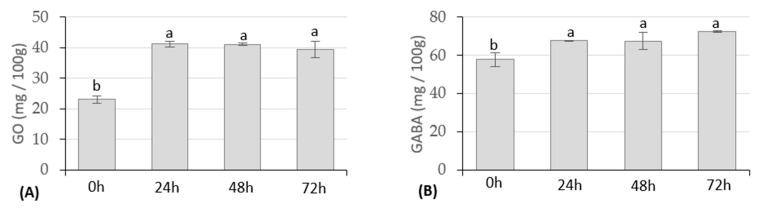
Effect of the germination time on the GO (**A**) and GABA (**B**) contents of brown Ariete variety rice, expressed in mg/100 g (DW). a, b—homogeneous groups according to Tukey’s post hoc test at a 95% confidence level.

**Figure 3 foods-12-01476-f003:**
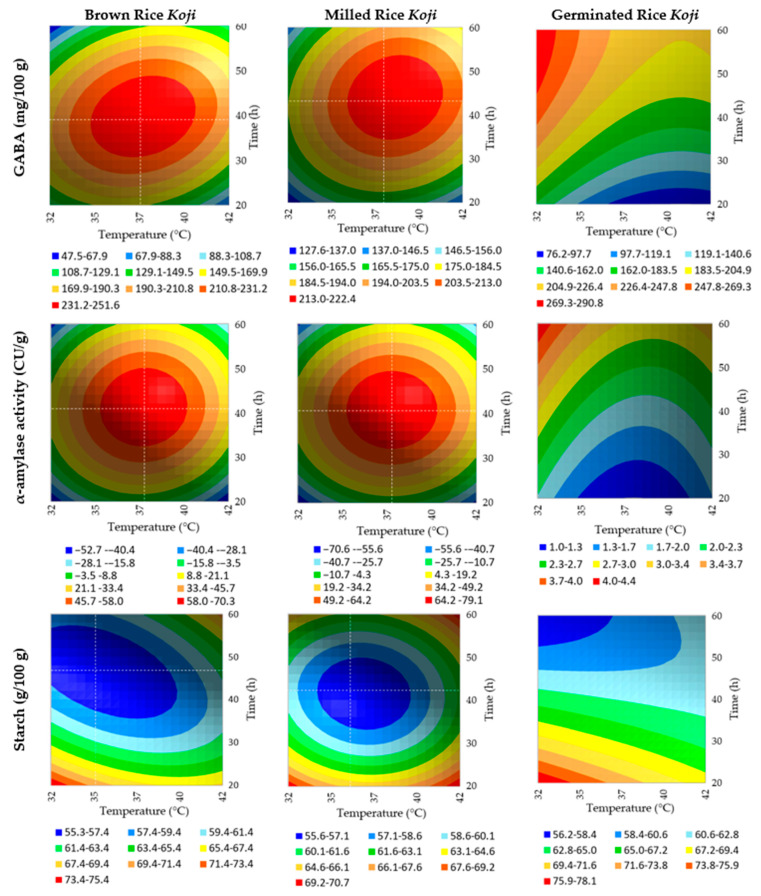
Contour plots of the predicted model for GABA, α-amylase activity and starch on brown, milled and germinated rice *Koji*. In GABA and α-amylase activity plots, white lines represent the temperature (°C) and time (h) targets to obtain the maximum response for GABA and α-amylase, and the minimum response for starch contents.

**Figure 4 foods-12-01476-f004:**
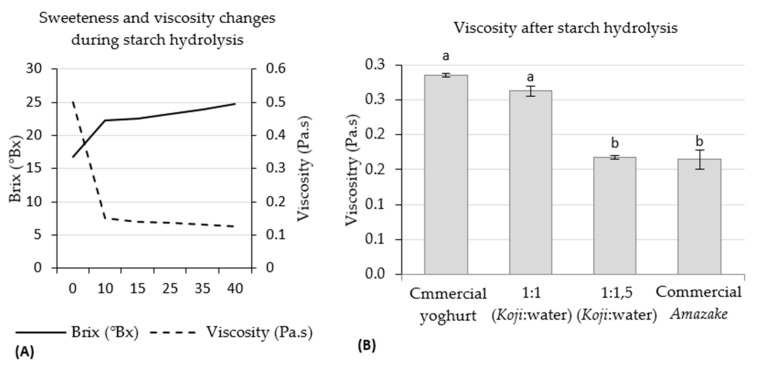
(**A**) Sweetness and viscosity over starch hydrolysis time. (**B**) Viscosity after hydrolysis in the selected *Koji*:water concentration and its comparison with drinking yoghurt and *Amazake*. a, b—homogeneous groups according to the Tukey’s post hoc test at a 95% confidence level.

**Figure 5 foods-12-01476-f005:**
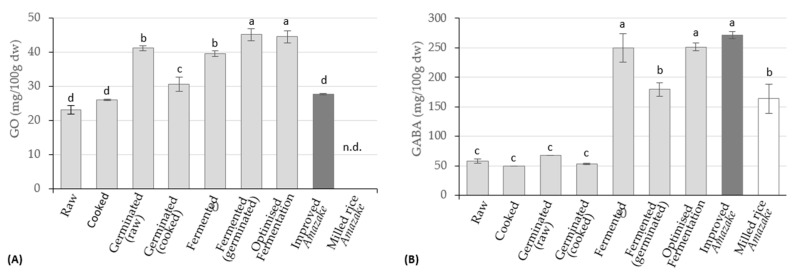
Effect of the various procedures on brown rice on (**A**) GO and (**B**) GABA contents. (n.d. stands for non-detectable). a, b, c, d—homogeneous groups according to the Tukey’s post hoc test at a 95% confidence level. The improved *Amazake* is shown in dark grey and the traditional milled rice *Anazake* in shown in white.

**Figure 6 foods-12-01476-f006:**
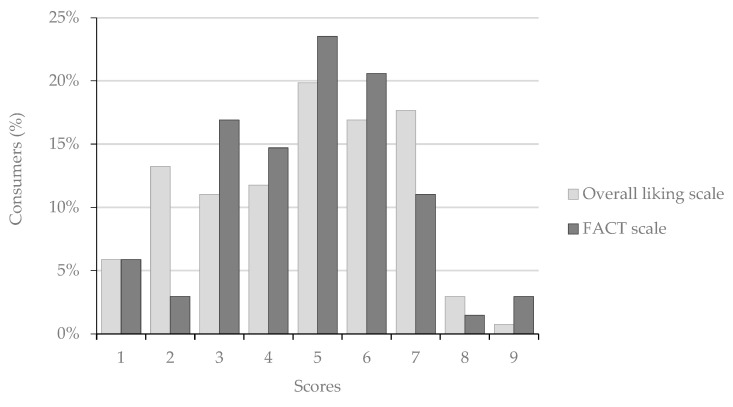
Distribution of consumers’ responses regarding overall liking and acceptance (FACT).

**Table 1 foods-12-01476-t001:** Definition of the circumscribed central composite design independent variable values. The α value related to ±1 is equal to ±1.414.

	Time (h)	Temperature (°C)
Min (−α)	20	32
Max (+α)	60	42
Centre (0)	40	37
Factorial (−1)	26	34
Factorial (+1)	54	41

**Table 2 foods-12-01476-t002:** Significant stepwise regression coefficients (±Std Error) (*p* < 0.05) and R^2^_adj_, from the fitting of Equation (1) to GABA concentration, α-amylase activity and starch content. Values for temperature (*x*_1_) were expressed in °C and incubation time (*x*_2_) was expressed in hours. Non-significant values (*p* > 0.05) are presented as n.s.

	b_0_	b_1_	b_2_	b_3_	b_4_	b_5_	AIC	R^2^_adj_
GABA (mg/100 g)								
Brown Rice	−3120 ± 777	171 ± 39.5	n.s.	−2.47 ± 0.524	0.307 ± 0.143	−0.255 ± 0.033	197	0.679
Milled Rice	−1811 ± 387	96.7 ± 20.2	8.92 ± 1.38	−1.27 ± 0.271	n.s.	−0.102 ± 0.017	151	0.681
Germinated Rice	1900 ± 971	−98.1 ± 50.8	9.12 ± 3.48	1.21 ± 0.681	n.s.	−0.073 ± 0.043	212	0.715
α-amylase activity (CU/g)								
Brown Rice	−3061 ± 221	153 ± 11.6	13.1 ± 0.815	−2.06 ± 0.155	n.s.	−0.159 ± 0.01	115	0.922
Milled Rice	−3490 ± 237	173 ± 12.5	16.2 ± 0.88	−2.33 ± 0.168	n.s.	−0.199 ± 0.01	114	0.939
Germinated Rice	44.0 ± 15.2	−2.31 ± 0.826	0.049 ± 0.008	0.03 ± 0.011	n.s.	n.s.	−29.3	0.618
Starch (g/100 g)								
Brown Rice	350 ± 105	−11.9 ± 5.3	−3.66 ± 0.80	n.s.	0.048 ± 0.019	0.021 ± 0.004	74.4	0.577
Milled Rice	360 ± 75.9	−14.7 ± 3.97	−1.76 ± 0.273	0.202 ± 0.053	n.s.	0.021 ± 0.003	57.3	0.622
Germinated Rice	153 ± 16.8	−1.75 ± 0.442	−2.38 ± 0.441	n.s.	0.038 ± 0.01	0.007 ± 0.002	38.1	0.876

**Table 3 foods-12-01476-t003:** Predicted maximum GABA and α-amylase activity according to the simultaneous linear optimisation of both adjusted quadratic models (Table 2).

	Temperature (°C)	Time (h)	GABA (mg/100 g)	α-Amylase Activity (CU/g)
Brown rice	37.2	40.3	251.6	70.4
Milled Rice	37.5	41.8	221.6	79.0
Germinated rice	32.0 *	60.0 *	290.5	4.0

* Within the experimental range.

## Data Availability

Data is contained within the article or Appendix A.

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
