# Peer review of "Improving γ-Oryzanol and γ-Aminobutyric Acid Contents in Rice Beverage Amazake Produced with Brown, Milled and Germinated Rices"

_foods, 2023, doi:10.3390/foods12071476_

Round 1

Reviewer 1 Report

Manuscript entitled ''Improving γ-oryzanol and γ-aminobutyric acid contents of rice beverage ‘Amazake’ optimised through germination and fermentation processes'' is an excellent study in which a product is developed from rice that is the staple food of Portugal. The introduction is too long. Formatting of the text is not done properly. And some other comments have been provided in the attached PDF.  

Author Response

Manuscript ID: foods-2184312

Title: Improving γ-oryzanol and γ-aminobutyric acid contents of rice beverage ‘Amazake’ optimised through germination and fermentation processes

Authors: Ana Castanho, Cristiana Pereira, Manuela Lageiro, Jorge C. Oliveira,
Luís Miguel Cunha, Carla Brites *

Answers to Reviewer #1

The authors thank the reviewer’s comments, which helped greatly improve the manuscript. Significant changes were performed: following the suggestions of reviewer one, the text was revised to improve clarity, and the flow chart was adjusted to correspond to the text sections to allow easier reading. Minor changes were also made throughout the paper, to improve English usage and grammar.

As the manuscript suffered significant modifications, the authors attach both versions of the revised manuscript, with and without track changes. The answer to each of the reviewer’s comments and questions follows below.

Thank you again for your contribution!

Reviewer 1

  1. Authors must correct the abstract to make it more technical. Otherwise it looks like a passage.
  2. Repeated sentence in abstract. line 13. sentence need to be replaced.

The abstract was revised following the reviewer's comment and considering the maximum word count required.

  1. L34 This paragraph should be joined together.

The authors fixed the issue following the reviewer's comment.

  1. Introduction is too long. Authors should make every paragraph connected with each other.

The authors thank the reviewer’s comment and have reviewed the Introduction section accordingly.

  1. The beverage is region specific. How it will be important for the rest of the world

Despite Amazake being a traditional beverage from Japan, it presents great potential for the rest of the rice-producer countries, as the fermentation process increases bioactive compounds and improves the nutritional value of rice. Additionally, the rice grains currently discarded for aesthetic reasons (e.g., broken grains) can be used to produce fermented beverages. Therefore, this process presents a high added value that deserves to be explored within different cultures, particularly for European consumers as they demand healthier and more sustainable food products. This idea is expressed in the introduction, which has been rewritten for the sake of clarity, and in the discussion section.

  1. Please explain some information about Rice Koji in the introduction??

The term "Rice Koji" was used since Koji can be made of other food matrices, such as beans, provided they are fermented with A. oryzae. This has been rewritten for the sake of clarity and the authors emphasised that Koji is the fermented rice that serves as a starter for Amazake.

  1. The formatting of this manuscript is totally scattrered into individual sentences. Please reformat the whole text. Results and discussion must be revised for better presentation of experimental findings.

The authors thank the comments and have thoroughly revised the manuscript to improve readability and to better explore the research findings.

  1. What conditions were optimised for germination and fermentaion?

Following the reviewers’ comments, the authors have realised that the method was not clear and that some misunderstanding was generated as only the Koji production was optimised. To improve interpretability, the authors have changed the title and have thoroughly revised the manuscript regarding the optimisation concept use.

  1. Please write about some future work. How to make it acceptable on sensorial basis despite of having numerous health benefits.

The authors thank the comment and have updated the manuscript to include future work at the end. The manuscript now reads:

“Despite the nutritional improvement associated with the bioactive compounds obtained in the Amazake, when presented to the consumers, overall liking scores were somehow low, which can be attributed to the product novelty and low familiarity with the concept of fermented rice beverages for Portuguese consumers. Nevertheless, as there is room for improvement and even adaptation of the formulation to the Portuguese consumers’ habits, further sensory improvement may be seek in the future with different variants of the fermented beverage, as well as the evaluation of acceptance within different consumption contexts.”

Reviewer 2 Report

The manuscript titled "Improving γ-oryzanol and γ-aminobutyric acid contents of rice beverage 'Amazake' optimized through germination and fermentation processes" mentions aiming to develop a rice-based beverage (Amazake) by germination and fermentation processes with A. Oryzae. As both GABA and GO can be naturally enhanced through biochemical processes, this paper explores ways to optimize the GABA of Amazake and GO, improving the nutritional value of Amazake and investigating its acceptability in the Portuguese market.

The manuscript lacks clarity and precision, and I list my observations below:

1. The title should change since only the fermentation stage was optimized, or at least it seems that way when reading the manuscript.

2. The article lacks clarity, especially it is not clear whether both the Koji preparation stage and the fermentation process were optimized, since the optimization designs in each case are not presented, and the contour plots of the surfaces of response for each stage, the characterization of the response surface, the mathematical models obtained with their corresponding adjustments, the optimal predicted values, as well as model validation experiments with the calculation of the error percentage for each one of them that has been demonstrated demonstrated a good fit to the second order model. For each stage where it is mentioned that it was optimized, the data described above must be presented.

3. In the materials and methods section on line 191 where optimization experiments are mentioned, I suggest including the mention of table 1 where it mentions the experimental conditions and specifying which stage it refers to (germination?, fermentation?). In fact, Table 1 is not mentioned in the text of the article.

4. The measurement of °Brix as an indicator of the amount of sugars is inappropriate for a scientific article to be published in this journal, since it is a very indirect measure of the sugar content. It is necessary to carry out more precise analyzes such as liquid chromatography or a minimum of spectrophotometry.

5. Subsection 2.5 “Amazake optimization”, change it to Amazake preparation, since in this section an optimization was not carried out, it only mentions that the optimized Koji conditions were used.

6. In the results and discussion section, in lines 295-296 they mention “a previous study was conducted to select the optimum germination time for increasing these compounds on brown rice”. According to figure 2 (line 298) optimization was not performed at this stage, only three germination times were tested and in fact in the materials and methods section (line 142) they clearly indicate "and the sample with greater values of those compounds were selected for further procedures.” Remember that in order to affirm that an optimization of conditions was carried out, the response surface methodology (RSM) must be used, which was not carried out in this germination stage. Given this situation, I recommend eliminating the term “optimum” when RSM had not been used, and replacing it with something like “the best conditions determined in relation to the highest values obtained for GO and GABA.

7. Lines 347-349 it is necessary to discuss the results shown in table 2. The regression coefficients are low, only GABA for germinated rice (R2 adj 0.710) and starch for the same raw material (R2 adj 0.928) they can be considered with adjustments for a second degree model and therefore as optimized responses

8. Similar to the previous comment (Lines 347-349) the discussion of figure 3 of the contours of the response surface is necessary. First, characterize the surface obtained for each case (maximum response point, minimum response point, saddle) to which the contours shown correspond. In my point of view, only the contours of the answers that were optimized should be shown in the article (there were only two, taking into account what is generally established that the minimum to consider that there is an adjustment to a model is 70%). . It is also not clear what are the optimal values that the model predicts and what is the response to these values.

9. Table 3 is not clear to me, they seem to be the same individual values for GABA (A) and α-amylase activity (B) or too similar, put back in (C) where they are supposed to correspond to GABA and α-amylase activity according to the adjusted model prediction. In addition, the models did not have a good fit as I mentioned above, except for two responses (GABA for germinated rice (R2 adj 0.710) and starch for the same raw material, that is, germinated rice (R2 adj 0.928) so this table shouldn't show up.

10. Lines 415-418 to be able to comment on this, they would need to show the standard deviation, percentage error of the model and mention how many replicates were carried out for the validation experiments, if they were carried out.

11. In section 3.4. Amazake optimization according to sweetness and viscosity, an optimization was not actually carried out, as I commented before they seem to interchangeably handle the optimization term with experiments where they tried various conditions and selected where it gave them the highest response. In order to call it an optimization they should have used RSM, otherwise it wasn't an optimization. They would have to correct throughout.

Author Response

Manuscript ID: foods-2184312

Title: Improving γ-oryzanol and γ-aminobutyric acid contents of rice beverage ‘Amazake’ optimised through germination and fermentation processes

Authors: Ana Castanho, Cristiana Pereira, Manuela Lageiro, Jorge C. Oliveira,
Luís Miguel Cunha, Carla Brites *

Answers to Reviewer #2

Reviewer 2

The authors thank the reviewer’s comments, which helped greatly improve the manuscript. Significant changes were performed, following the suggestions of reviewer two. The raw data was all verified and is now available in supplementary materials for the sake of transparency; the title was modified according to the suggestions of the reviewer, and the RSM models were refitted. the text was revised to improve clarity, and the flow chart was adjusted to correspond to the text sections to allow easier reading. Minor changes were also made throughout the paper, to improve English usage and grammar.

As the manuscript suffered significant modifications, the authors attach both versions of the revised manuscript, with and without track changes. The answer to each of the reviewer’s comments and questions follows below.

Thank you again for your contribution!

The manuscript titled "Improving γ-oryzanol and γ-aminobutyric acid contents of rice beverage 'Amazake' optimised through germination and fermentation processes" mentions aiming to develop a rice-based beverage (Amazake) by germination and fermentation processes with A. Oryzae. As both GABA and GO can be naturally enhanced through biochemical processes, this paper explores ways to optimise the GABA of Amazake and GO, improving the nutritional value of Amazake and investigating its acceptability in the Portuguese market.

The manuscript lacks clarity and precision, and I list my observations below:

  1. The title should change since only the fermentation stage was optimised, or at least it seems that way when reading the manuscript.
  2. The article lacks clarity, especially it is not clear whether both the Koji preparation stage and the fermentation process were optimised, since the optimisation designs in each case are not presented, and the contour plots of the surfaces of response for each stage, the characterisation of the response surface, the mathematical models obtained with their corresponding adjustments, the optimal predicted values, as well as model validation experiments with the calculation of the error percentage for each one of them that has been demonstrated a good fit to the second order model. For each stage where it is mentioned that it was optimised, the data described above must be presented.
  3. In the materials and methods section on line 191 where optimisation experiments are mentioned, I suggest including the mention of table 1 where it mentions the experimental conditions and specifying which stage it refers to (germination?, fermentation?). In fact, Table 1 is not mentioned in the text of the article.

The authors thank the reviewer’s comments and confirm that optimisation was only applied to the rice Koji fermentation process. The manuscript was thoroughly revised to clarify this and the title was modified accordingly. Additionally, the term “optimisation” was only used when referring to rice Koji fermentation. The authors emphasised that Koji is the result of rice fermentation and that this stage was the only one to be optimised. The Materials and Methods section was modified to clearly present the Koji optimisation model design and conditions. The authors also included the standard errors for each model parameter presented in the Koji optimisation process.

The authors solved the issue regarding Table 1 and restructured the paper to correspond to the stages presented on the flowchart (Figure 1), to improve readability.

  1. The measurement of °Brix as an indicator of the amount of sugars is inappropriate for a scientific article to be published in this journal, since it is a very indirect measure of the sugar content. It is necessary to carry out more precise analyses such as liquid chromatography or a minimum of spectrophotometry.

The authors agree and add a phrase in the paper regarding the choice of °Brix to measure sweetness. As it is expressed in the paper, the °Brix was used as an indicator and an expedited measure to better control the hydrolysis process in real-time, thus avoiding extending excessively the scarification process. Moreover, the authors consider that the sweetness measurement was secondary to the experiment objective.

  1. Subsection 2.5 "Amazake optimisation", change it to Amazake preparation, since in this section an optimisation was not carried out, it only mentions that the optimised Koji conditions were used.
  2. In the results and discussion section, in lines 295-296 they mention "a previous study was conducted to select the optimum germination time for increasing these compounds on brown rice". According to figure 2 (line 298) optimisation was not performed at this stage, only three germination times were tested and in fact in the materials and methods section (line 142) they clearly indicate "and the sample with greater values of those compounds were selected for further procedures." Remember that in order to affirm that an optimisation of conditions was carried out, the response surface methodology (RSM) must be used, which was not carried out in this germination stage. Given this situation, I recommend eliminating the term "optimum" when RSM had not been used, and replacing it with something like "the best conditions determined in relation to the highest values obtained for GO and GABA.

As referred previously, the authors agree and have revised the manuscript accordingly to better express this, considering that only the rice Koji process was optimised.

  1. Lines 347-349 it is necessary to discuss the results shown in table 2. The regression coefficients are low, only GABA for germinated rice (R2 adj 0.710) and starch for the same raw material (R2 adj 0.928) they can be considered with adjustments for a second degree model and therefore as optimised responses

The authors agree that the values are lower than expected and following the reviewer’s comment have decided to revise the raw experimental data. The samples were collected as independent triplicates due to the complexity of the experiment. The implementation of the different processing steps and the nature of the rice fermentation process may have contributed to higher variability of the data, thus affecting the model fitting process. A closer inspection of the data has revealed the presence of a couple of outliers. After removing them, following the Grubbs method and guaranteeing that the Relative Standard Deviation for each experimental point was below 20 % (ref), the quadratic model from Eq. (1) was fitted to the data. The model building process was further optimised by minimising Aikaike’s AIC value.

  1. Similar to the previous comment (Lines 347-349) the discussion of figure 3 of the contours of the response surface is necessary. First, characterise the surface obtained for each case (maximum response point, minimum response point, saddle) to which the contours shown correspond. In my point of view, only the contours of the answers that were optimised should be shown in the article (there were only two, taking into account what is generally established that the minimum to consider that there is an adjustment to a model is 70%). . It is also not clear what are the optimal values that the model predicts and what is the response to these values.
  2. Table 3 is not clear to me, they seem to be the same individual values for GABA (A) and α-amylase activity (B) or too similar, put back in (C) where they are supposed to correspond to GABA and α-amylase activity according to the adjusted model prediction. In addition, the models did not have a good fit as I mentioned above, except for two responses (GABA for germinated rice (R2 adj 0.710) and starch for the same raw material, that is, germinated rice (R2 adj 0.928) so this table shouldn't show up.

Indeed, the Radj2 values were lower than expected, as referred to before. The lines indicating the lower value were added and the table containing the predictions was simplified to only include the conjoint predicted maximum. As the whole experiment was carried out in stages, focusing on the best result regarding GO and GABA, the response values were only obtained for germinated rice Koji, as it was suggested to have the higher results, and were not considered in the table but in the text immediately after. Table 2 was modified, following a full revision of the raw data points and using the Aikaike’s AIC- value as the objective function for the model-building process, following a stepwise regression approach. Fortunately, this has strongly improved the Radj2 values, thus the prediction power of the final models. The optimisation conditions presented in Table 3, as well as in Figure 3, have also been reviewed accordingly. Table 3 was also restructured to avoid doubled information and for clarity.

  1. Lines 415-418 to be able to comment on this, they would need to show the standard deviation, percentage error of the model and mention how many replicates were carried out for the validation experiments, if they were carried out.

Thank you for your comment. The standard error was included in the table, and the mention to the replicates was indeed missing; it was added in the materials and methods section.

  1. In section 3.4. Amazake optimisation according to sweetness and viscosity, an optimisation was not actually carried out, as I commented before they seem to interchangeably handle the optimisation term with experiments where they tried various conditions and selected where it gave them the highest response. In order to call it an optimisation they should have used RSM, otherwise it wasn't an optimisation. They would have to correct throughout.

The full manuscript has suffered deep changes regarding the use of the term optimisation. The authors would like to thank the time the reviewer spent to read and comment the manuscript, and all the helpful comments regarding this particular issue.

Round 2

Reviewer 2 Report

The article has been substantially improved, making it clearer and easier to read. I only have a couple of comments:

1. In table 1 it seems to me that there is an error in the values that correspond to (-α), (α) and those that correspond to (-1) and (1) are inverted

2. Figure 3 is not yet explained and discussed in the text of the article, it would be convenient to add it so as not to limit itself to putting it in the manuscript.

Author Response

Dear reviewer,

The authors would like to thank your comments. The responses are in the attached file.

Best Regards
